# Single-Molecule Super-Resolution Microscopy Reveals Heteromeric Complexes of MET and EGFR upon Ligand Activation

**DOI:** 10.3390/ijms21082803

**Published:** 2020-04-17

**Authors:** Marie-Lena I.E. Harwardt, Mark S. Schröder, Yunqing Li, Sebastian Malkusch, Petra Freund, Shashi Gupta, Nebojsa Janjic, Sebastian Strauss, Ralf Jungmann, Marina S. Dietz, Mike Heilemann

**Affiliations:** 1Single Molecule Biophysics, Institute of Physical and Theoretical Chemistry, Goethe University Frankfurt, Max-von-Laue-Str. 7, 60438 Frankfurt, Germany; 2SomaLogic, Inc., Boulder, CO 80301, USA; 3Department of Physics and Center for Nanoscience, Ludwig Maximilian University, 80539 Munich, Germany; 4Max Planck Institute of Biochemistry, 82152 Planegg, Germany

**Keywords:** receptor tyrosine kinases, MET, EGFR, receptor cross-interaction, single-molecule localization microscopy, single-particle tracking, DNA-PAINT

## Abstract

Receptor tyrosine kinases (RTKs) orchestrate cell motility and differentiation. Deregulated RTKs may promote cancer and are prime targets for specific inhibitors. Increasing evidence indicates that resistance to inhibitor treatment involves receptor cross-interactions circumventing inhibition of one RTK by activating alternative signaling pathways. Here, we used single-molecule super-resolution microscopy to simultaneously visualize single MET and epidermal growth factor receptor (EGFR) clusters in two cancer cell lines, HeLa and BT-20, in fixed and living cells. We found heteromeric receptor clusters of EGFR and MET in both cell types, promoted by ligand activation. Single-protein tracking experiments in living cells revealed that both MET and EGFR respond to their cognate as well as non-cognate ligands by slower diffusion. In summary, for the first time, we present static as well as dynamic evidence of the presence of heteromeric clusters of MET and EGFR on the cell membrane that correlates with the relative surface expression levels of the two receptors.

## 1. Introduction

Receptor tyrosine kinases (RTKs) form a multifaceted family of transmembrane receptors. They orchestrate cellular communication, growth, motility, and differentiation [1,2]. Most RTKs undergo ligand-induced dimerization, trans-phosphorylation by the kinase domain, and initiate downstream signaling [3,4]. Overexpression or mutations of RTKs figure prominently in the development of several diseases, such as inflammation, diabetes, or cancer. This fact renders the RTK family a focus of pharmaceutical therapies [5,6].

The MET receptor and the epidermal growth factor receptor (EGFR) are two out of 58 known human RTKs embodying members of two different receptor subfamilies [2]. They are bound and activated by their physiological ligands hepatocyte growth factor (HGF) and epidermal growth factor (EGF), respectively. Both receptors play a role in the development and growth of various types of malignant carcinomas by excessive signaling that contributes to tumor proliferation. MET or EGFR mutations are detected in 3–30% of non-small-cell lung carcinoma (NSCLC) and are responsible for decreased receptor degradation as well as overexpression resulting in increased signaling [7,8].

In the course of cancer therapy, MET and EGFR signaling pathways are often targeted either by monoclonal antibodies to prevent binding of activating ligands or by small-molecule inhibitors of receptor phosphorylation [9,10]. Nowadays, cancer patients are routinely treated with therapeutic agents such as erlotinib and gefitinib to suppress EGFR signaling [11,12] or crizotinib and cabozantinib targeting MET [13,14]. However, the success of these therapies is often limited by the emergence of tumor resistance against such inhibitors [15]. The occurrence of these resistances is partially attributed to further receptor mutations [16,17]. Still, the inhibitor resistance in a significant number of cases cannot be explained satisfactorily by any kind of mutation. Instead, increasing evidence points to RTK interaction across subfamilies, thus circumventing inhibition of single receptors by hijacking alternative activation or signaling pathways [9,10,15]. The extent, complexity, and mechanisms of receptor cross-interactions purportedly responsible for inhibitor resistance are not fully understood. An increasing number of studies hints at cross-interaction of MET and EGFR following ligand stimulation. However, the literature on details of the phenomenon and underlying principles is very diverse, sometimes contradictory, and does not provide a comprehensive interpretation of MET and EGFR interaction. There are indications that EGF stimulation may lead to EGFR-MET interactions in breast cancer cells but no such effects were observed upon HGF activation [18]. Increased MET phosphorylation was reported in EGF-stimulated human epidermoid carcinoma [19]. Conversely, transactivation of EGFR by HGF-stimulated MET was observed during retinal pigment epithelial wound healing [20] as well as in mammary epithelial cells [21]. The possible formation of MET:EGFR heterodimers is discussed in this context [22].

The expression ratio of MET and EGFR has been discussed as a possible explanation for the occurrence of receptor interaction and subsequent inhibitor resistance in certain cell lines. For instance, transactivation of MET coincides with increased expression of EGFR in human epidermoid carcinoma [19]. Another study found enhanced receptor signaling upon EGFR activation coinciding with elevated MET expression in NSCLC [23]. Park et al. (2015) found that the success of EGFR inhibitors in NSCLC is dependent on MET:EGFR expression ratios.

Several studies have provided evidence of interactions between MET and EGFR upon ligand binding in several different cell types. This suggests that receptor cross-interactions are governed by a complex system of factors, including cell type, absolute and relative receptor expression levels, and ligand stimulation. Correlation of these factors in the bigger context of cancer therapy requires understanding the basic principles and governing patterns of transient receptor cross-interaction and possible quantification of the size of the effect depending on various cellular and environmental factors. Thus, it is crucial to develop systematic and highly sensitive methods to study EGFR and MET behavior upon ligand stimulation. In this work, we used single-molecule super-resolution microscopy to visualize single receptor clusters of EGFR and MET in two cancer cell lines. Exchange-PAINT (Point Accumulation for Imaging in Nanoscale Topography) [24,25] was applied to analyze and quantify EGFR and MET receptor densities and colocalization in situ. Receptor dynamics in living cells were measured by single-particle tracking [26] and stimulation by HGF as well as EGF. A breast cancer cell line (BT-20) and cervical cancer cells (HeLa) were selected due to distinctly different MET:EGFR expression levels. For the first time, we detected heteromeric clusters of MET and EGFR directly in cell membranes.

## 2. Results

We introduce a quantitative, imaging-based approach to identify cross-interactions between MET and EGFR in HeLa and BT-20 cells, which exhibit distinctly different receptor expression ratios [27,28]. Using single-molecule super-resolution microscopy (Figure 1a) in combination with single-molecule colocalization analysis, we visualized single receptor complexes with nanoscale resolution in cells and found direct proof of in situ formation of heteromeric complexes of MET and EGFR.

### 2.1. Membrane Receptor Densities of MET and EGFR Are Influenced by HGF as Well as EGF Stimulation

We visualized single receptor clusters of EGFR and MET in the cellular plasma membrane using multiplexed single-molecule super-resolution microscopy. We used Exchange-PAINT in combination with immunofluorescence and DNA-labeled secondary antibodies to visualize both receptors in the same cell (Figure 1a) [29].

MET and EGFR were imaged in HeLa as well as BT-20 cells, either unstimulated or stimulated by HGF or EGF. Varying receptor cluster densities depending on cell type and ligand stimulation were visible in super-resolution images (Figure 1b). We analyzed the DNA-PAINT images with DBSCAN (density-based spatial clustering and application with noise) [30] to obtain average receptor cluster densities following ligand stimulation, which are shown along with a schematic illustration of changes on cell surfaces for both HeLa (Figure 1c, Appendix A) and BT-20 cells (Figure 1d, Appendix A). In unstimulated HeLa cells, MET is about two-fold more abundant on the cell surface compared to EGFR (14.1 ± 0.5 MET receptors/µm^2^ and 6.3 ± 1.5 EGFR/µm^2^). Upon activation with HGF, the number of MET receptors on the cell surface dropped by 2.2-fold in a highly significant manner (*p*-value = 4 × 10^−7^) while the EGFR surface density increased slightly (*p*-value = 0.040). In EGF-stimulated HeLa cells, MET density decreased significantly by 1.1-fold (*p*-value = 0.031) while the EGFR density on the cell membrane decreased 4.2-fold in a highly significant manner (*p*-value = 6 × 10^−6^).

In unstimulated BT-20 cells, which express similar levels of MET but considerably higher levels of EGFR compared with HeLa cells, we found a nearly inverse MET:EGFR ratio with 11.8 ± 1.5 MET/µm^2^ and 20.8 ± 1.6 EGFR/µm^2^. In HGF-stimulated cells, MET surface densities dropped by 1.4-fold (*p*-value = 0.007) while EGFR showed little change (1.1-fold decrease, *p*-value = 0.303). EGF activation of BT-20 cells caused a highly significant 2.6-fold reduction of EGFR on the cell surface (*p*-value = 1 × 10^−6^) similar to the effect in HeLa cells. The MET surface density exhibited a nearly significant decrease of 1.2-fold (*p*-value = 0.056) compared to resting cells.

### 2.2. Colocalization between MET and EGFR Increases upon Ligand Activation

In order to detect heteromeric complexes of EGFR and MET, DNA-PAINT data were processed with the coordinate-based colocalization (CBC) analysis [31] using the software tool LocAlization Microscopy Analyzer (LAMA, version 16.10) [32]. Within a distance of 30 nm, we determined the degree of colocalization by calculating the CBC values that ranged between −1 and 1, where positive values indicate the colocalization of both receptors. The CBC analysis can distinguish between EGFR colocalizing with MET, and MET colocalizing with EGFR (Figure 2a), and was performed for both scenarios. We next filtered our data for positive CBC values (>0.15) and generated super-resolved colocalization images that clearly show MET and EGFR forming heteromeric MET:EGFR complexes (Figure 2b). The degree of colocalization was determined from the number of colocalized MET:EGFR clusters compared to the total number of receptor clusters (Figure 2c). To highlight the membrane densities of EGFR and MET, as well as of heteromeric MET:EGFR, we constructed Venn diagrams for HeLa and BT-20 cells (Figure 2d).

Receptor colocalization was analyzed in unstimulated and stimulated cells. In HeLa cells, the relative amount of MET colocalizing with EGFR increased by 2.3-fold upon HGF activation compared to resting cells (*p*-value = 6 × 10^−5^) but dropped by 1.6-fold upon EGF stimulation (*p*-value = 0.034). In BT-20 cells, relative colocalization levels were generally higher than in HeLa cells. While stimulation with HGF had no effect on the relative amount of MET clusters colocalizing with EGFR compared to the unstimulated condition (*p*-value = 0.811), EGF stimulation resulted in a 2.2-fold decrease in colocalization (*p*-value = 0.002). For EGFR colocalizing with MET in HeLa cells, stimulation with HGF led to a 1.3-fold decrease in colocalization (*p*-value = 0.047), whereas stimulation with EGF led to a 2.2-fold increase in colocalization (*p*-value = 2 × 10^−4^). In BT-20 cells, the frequency of EGFR colocalizing with MET increases by 1.1-fold upon HGF stimulation, but this change is not significant (*p*-value = 0.616) (Figure 2c,d). As for all other conditions, colocalization increased significantly upon stimulation with the physiological ligand, in this case, EGF (1.3-fold increase, *p*-value = 0.002). Venn diagrams highlight that the relative number of colocalizing receptors stays high even upon reduction of receptor surface densities, e.g., upon HGF stimulation.

### 2.3. Ligand Binding Reduces the Diffusion Coefficients of MET and EGFR in Living Cells

We measured how ligand stimulation affects diffusion of EGFR and MET in living cells with single-particle tracking (SPT). MET was targeted with a fluorescently labeled antibody fragment, Fab-ATTO 647N (Figure 3a). EGFR was targeted by a DNA-based slow off-rate modified aptamer (SOMAmer reagent) [33] connected to a P1 DNA docking strand which was transiently bound by the complementary imager strand labeled with Cy3B (Figure 3d). All SPT experiments were performed in unstimulated cells as well as ligand-stimulated cells (Figure 3a,d). Average diffusion coefficients were determined relative to unstimulated cells. All given *p*-values from two-sample t-tests were supported by a graphical depiction of the distributions of mean differences of each condition compared to resting cells together with the respective 95% confidence interval which visualizes the heterogeneity and differences of the tested conditions (Figure 3b,c,e,f) [34]. The relative frequencies of diffusion types were identified and depicted as bar plots (Appendix A).

We first examined whether the fluorophore-labeled ligands Fab-ATTO 647N and EGFR-SOMAmer reagent interfere with or enhance the activation of the receptors by their physiological ligands HGF and EGF. We used receptor and phosphorylated-receptor specific antibodies to identify total and active MET and EGFR in cells (Appendix A). Blots of unstimulated HeLa and BT-20 cells showed no activation of MET or EGFR by these fluorophore-labeled ligands. Blots of cells in the resting state or stimulated with HGF and EGF showed the same levels of phosphorylated MET and EGFR in the presence and absence of labels, indicating that cognate ligand addition activates MET and EGFR, respectively, and receptor activation was not stimulated or hampered by the presence of fluorescent ligands. Thus, we conclude that the labels themselves do not interfere with receptor activation.

We next determined the optimal ligand incubation time by monitoring the effects of long-term EGF incubation on the diffusion dynamics of MET (Appendix A). After 15 min of EGF stimulation, the normalized mean diffusion coefficient decreased to 93% of the mean value in resting cells. After six hours, this value dropped to 82%, and after 48 h to 74%. However, when scrutinizing cell morphology (Appendix A), we found a substantial fraction of cells were rounded up. We, therefore, chose to perform all SPT experiments directly after ligand stimulation.

In HeLa cells treated with HGF, we observed that the average diffusion coefficient of MET dropped highly significantly compared to unstimulated cells (*p*-value = 5 × 10^−12^) (Figure 3b and Appendix A). Diffusion type analysis also showed decreased diffusion coefficients of the free and confined population as well as an increase of the fraction of immobile receptors. Stimulation by EGF also led to a slowing down of MET relative to resting cells representing a very significant change (*p*-value = 0.006). In this case, there is also a subtle decrease in diffusion coefficients of the free and confined population as well as a slightly increasing immobile population.

To exclude the possibility that receptor mobility is influenced by changes in the biophysical properties of the plasma membrane, we performed single-molecule tracking experiments using a transmembrane domain (TMD) fused to a monomeric enhanced green fluorescent protein (mEGFP) [35] (Appendix A). In HeLa cells that were transiently expressing mEGFP-TMD, we measured the mobility of mEGFP-TMD using an anti-GFP nanobody labeled with Abberior Star 635P, and of MET using the Fab-ATTO 647N. We did not observe any change in diffusion of TMD upon cell stimulation with HGF or EGF (Appendix A), while MET diffusion showed the same trends upon ligand treatment as in cells that were not transfected with mEGFP-TMD.

Next, MET diffusion dynamics were studied in BT-20 cells, which showed a more heterogeneous distribution than in HeLa cells (Figure 3c and Appendix A). Upon activation with HGF, the diffusion coefficient was reduced significantly compared to resting cells (*p*-value = 0.024). The mean diffusion coefficients of both mobile fractions were slightly decreased while the portion of immobile particles increased distinctly. Upon EGF treatment of cells, MET diffusion was slowed compared to resting cells. However, this does not constitute a significant difference (*p*-value = 0.173) although there is also a slightly increased immobile portion when considering the diffusion type analysis.

In HeLa cells stimulated with EGF, EGFR yielded highly significantly lower diffusion coefficients in comparison to unstimulated cells (*p*-value = 3 × 10^−4^) (Figure 3e and Appendix A). Diffusion type analysis exhibited slightly decreased mean diffusion coefficients for both mobile populations and an increase in the populations of immobile and confined receptors. SPT experiments in HGF-stimulated cells yielded a very significantly reduced EGFR diffusion coefficient in contrast to the value in unstimulated cells (*p*-value = 0.003). Changes in the mean diffusion coefficients of confined and free receptors were very comparable to those observed for EGF activation, the portion of immobile particles was also clearly amplified.

Finally, EGFR diffusion dynamics were also monitored in BT-20 cells (Figure 3f and Appendix A). As for MET, the distributions of mean diffusion coefficients per cell as well as of mean differences between conditions are broader than in HeLa cells. Regarding the effects of EGF stimulation, activation by the physiological ligand resulted in a diffusion coefficient significantly reduced relative to unstimulated EGFR (*p*-value = 0.011). Diffusion type analysis revealed a slightly reduced mean diffusion coefficient of freely diffusing proteins and a larger population of immobile receptors. The addition of HGF yielded a very significant attenuation of the overall mean diffusion coefficient compared to resting cells (*p*-value = 0.002). Diffusion type analysis showed similar results for mean diffusion coefficients and diffusion types as for EGF-stimulated cells.

### 2.4. Western Blots do not Reveal Cross-Phosphorylation of EGFR and MET upon Ligand Stimulation

Western blots are a well-established method to detect proteins and modifications such as phosphorylation. We tested whether we could detect a ligand-induced interaction of MET and EGFR by monitoring phosphorylated receptors at two different time points (Appendix A). Band intensities of phosphorylated receptors were compared for unstimulated and stimulated cells. We did not observe any increase in phosphorylated EGFR or MET in stimulated HeLa (Appendix A) or BT-20 cells (Appendix A) due to potential cross-activation. Phospho-MET only increased above the level of basal activity upon stimulation with HGF. Phospho-EGFR increased for cells activated with EGF or both ligands. In summary, western blots did not report a cross-phosphorylation of EGFR or MET under the conditions tested.

## 3. Discussion

Weak and infrequent interactions of proteins often are not accessible with biochemical methods such as co-immunoprecipitation. Therefore, we established a systematic and highly sensitive single-molecule localization microscopy approach which provided direct proof of the formation of heteromeric complexes of MET and EGFR. We studied the interactions of MET and EGFR upon stimulation by HGF or EGF in the two cancer cell lines, HeLa and BT-20, which exhibit inverse MET:EGFR expression ratios. We identified and quantitatively evaluated the formation of heteromeric complexes in the context of ligand stimulation and receptor expression ratios.

### 3.1. Indications for Cross-Interactions of MET and EGFR upon Activation with HGF as Well as EGF

RTKs are dimerized upon ligand binding [3], internalized, and degraded or recycled as a main regulatory process upon activation [36]. This accounts for reduced receptor densities on cell membranes after activation with the physiological ligands for both MET and EGFR in HeLa as well as BT-20 cells (Figure 1c,d). The possibility of receptor interactions of MET upon activation of EGFR with EGF is supported by the observation that MET densities are reduced on the cell membrane in EGF-stimulated cells. The reduction in MET surface density upon EGF stimulation, observed in both HeLa cells (*p*-value = 0.031) and BT-20 cells (nearly significant, *p*-value = 0.056), is distinctly smaller compared to the effect of activation with the natural ligand, HGF. We want to note at this point that we think it is essential to evaluate statistics with care in the context of real-life systems and discuss *p*-values as trends in a broader context rather than assessing them as strict borders as has been debated critically in several recent studies [34,37]. While a portion of MET receptors seem to be internalized in both cell types after EGF stimulation, this effect was not observed for EGFR in cells stimulated with HGF in either cell type. Although there is a subtle decrease in EGFR densities in BT-20 stimulated with HGF, this effect is small and suffers from the high heterogeneity of EGFR densities on the plasma membrane of BT-20 cells. In conclusion, receptor density analysis suggests an effect of EGF stimulation on MET receptor while there is no evidence for an effect of HGF stimulation on EGFR.

Further evidence of interactions between MET and EGFR was gathered from colocalization analysis of super-resolved receptor clusters (Figure 2). Visualization and analysis of heteromeric MET:EGFR complexes was achieved. For MET, we observed a significant relative increase of colocalization with EGFR in HGF stimulated HeLa cells (Figure 2c). This indicates a raised relative level of MET:EGFR heteromeric complexes for active MET receptors still present at the cell membrane. Upon EGF stimulation, MET colocalization with EGFR decreased in both cell lines (Figure 2c). This is probably because EGFR levels at the membrane are distinctly reduced upon activation (Figure 1c,d, and Figure 2d), therefore reducing the probability of its interaction with MET.

For EGFR, we observed significant increases in its relative colocalization with MET in HeLa and BT-20 cells stimulated with EGF. Stimulation with HGF led to a decrease in the relative number of MET:EGFR complexes in HeLa cells, likely because of lower MET surface densities and thus a lower probability of “finding” an interaction partner at the cell membrane. BT-20 cells stimulated with EGF or HGF both showed a slight increase in EGFR colocalizing with MET when normalized to receptor membrane densities (Figure 2c). While we were not able to deduce the effects of HGF-stimulated MET on EGFR from receptor densities at the cell membrane (Figure 1d), this subtle rise of colocalization may suggest the formation of MET:EGFR heteromeric complexes in BT-20 cells treated with HGF. Generally, the relative fraction of heteromeric complexes increased upon activation by the respective physiological ligand and prior to internalization. The formation of heteromeric receptor complexes may either be due to an interaction between ligand-stimulated homodimers and the other RTK (cross-interactions) (Figure 2e) or due to direct heterodimerization (cross-activation) [22].

Tracking single receptors in living cells provides further dynamic evidence of cross-interaction between MET and EGFR. In a previous study, we showed that ligand activation decreased the diffusion coefficient of MET and increased the fraction of immobile receptors, possibly because of immobilization prior to endocytosis [38]. For EGFR in cells stimulated with EGF, similar observations were made [39,40]. Consistent with these studies, we observed reduced diffusion coefficients for both MET and EGFR in cells stimulated with their physiologic ligands, HGF and EGF, respectively (Figure 3, Appendix A). In addition, we found a decrease in the diffusion coefficient of MET in HeLa cells stimulated with EGF (Figure 3b). Together with a decrease in receptor densities seen in DNA-PAINT data, this observation further suggests an interaction of MET with EGFR and a possible increase in MET internalization in EGF-stimulated cells. A similar effect was not observed in BT-20 cells stimulated with EGF (Figure 3c). For EGFR, we observed very significant reduction in diffusion coefficients in both HeLa and BT-20 cells when stimulated with HGF (Figure 3e,f). Together with the relative increase in MET:EGFR complexes in HGF-treated cells, this supports the hypothesis of an interaction of MET with EGFR (Figure 2b), although EGFR densities remained unchanged (Figure 1c,d).

To exclude a potential, direct or non-direct, effect of HGF or EGF stimulation on membrane stiffness or cytoskeletal organization, which in turn might influence receptor diffusion dynamics, we performed a control experiment with mEGFP-TMD [35], which does not interact with EGFR or MET. We found that the mobility of mEGFP-TMD did not change significantly upon treatment of the cells with HGF or EGF (Appendix A). This further supports the hypothesis that the observed changes in diffusion of EGFR and MET may originate from a MET:EGFR interaction.

Taken together, we have been able to visualize and analyze the formation of heteromeric complexes of MET and EGFR in both HeLa and BT-20 cells stimulated either with EGF or HGF. These systematic findings are supported by reports of interactions of MET and EGFR or other members of these receptor subfamilies in the context of drug-targeting of these RTKs [15,18,19,20]. Nevertheless, the molecular mechanism of this interaction remains elusive. There are reports on cross-phosphorylation (although we have not been able to confirm it in this study) and heterodimer formation [19,41], which may explain the slower diffusion, lower membrane densities, and thus supposedly increased internalization rates of MET or EGFR upon activation of the other receptor.

### 3.2. Occurrence and Strength of the Cross-Interaction Depend on Receptor Expression Ratios

For this study, we used two cancer cell lines with different receptor expression ratios. For HeLa cells, reported mRNA expression ratios range from approximately 4:1 to 6:1 MET:EGFR [42,43]; [27,28]. For BT-20 cells, mRNA expression ratios of 1:4 MET:EGFR are published [27,28]. mRNA expression levels can give a rough indication on membrane receptor densities [44] would estimate ratios of 1:1.6 MET:EGFR in HeLa cells and 1:23 in BT-20 cells. Our super-resolution data on receptor densities (Figure 1c,d) report ratios of approximately 2:1 MET:EGFR in HeLa cells and of 1:2 in BT-20 cells. Absolute numbers were around 13 MET receptors/µm^2^ in both cell lines, EGFR exhibited distinctly different densities of about 6 receptors/µm^2^ in HeLa and 21 receptors/µm^2^ in BT-20 cells. We note that receptor numbers represent receptor cluster numbers rather than absolute numbers of single receptors. Especially at high receptor densities as in the case of EGFR in BT-20 cells, we probably underestimate receptor numbers. Different receptor ratios have previously been discussed to influence the strength or occurrence of receptor interactions. Heterodimers and cross-phosphorylation preferentially occurred in lung cancer cells with amplified MET ratios (42). Park et al. (2015) reported that low MET:EGFR ratios correlated with resistance to EGFR inhibitors.

In our study, we were able to systematically evaluate the effect of receptor expression ratios on cross-interactions of MET and EGFR. Considering our DNA-PAINT data, we found no distinct differences in the way receptors reacted to ligand stimulation (Figure 1c,d). In both cell lines, we did not observe an effect of HGF stimulation on EGFR densities, but a slight influence of EGF stimulation on MET receptor densities. The decrease in MET density was greater in BT-20, which may be explained by the diverse MET:EGFR ratio rendering heteromeric formation or cross-activation and subsequent co-internalization of MET with the abundant active EGFR molecules more probable. Regarding co-localization analysis (Figure 2c,d), the occurrence of EGFR colocalizing with MET normalized against absolute receptor densities is slightly increased upon both HGF and EGF stimulation in BT-20 cells in contrast to all other plots only showing enhanced receptor colocalization upon activation with the respective physiological ligand but decreased relative colocalization levels upon addition of the other ligand. We hypothesize that due to the abundance of EGFR even the effect of MET internalization upon activation by HGF is not sufficient to overcome the effect of increased heteromeric complex formation of MET and EGFR. In SPT live-cell data, we observed more distinctly decreased mean diffusion coefficients for EGFR in HGF-stimulated BT-20 cells compared to HeLa cells (Figure 3e,f). This may support the BT-20 colocalization data where interactions or heterodimerization of activated MET with EGFR is probable due to the excess of EGFR. In contrast, EGF stimulation affected MET dynamics in HeLa but not in BT-20 cells. In SPT experiments, only a small fraction of receptors is observed due to the necessity of very sparse labeling. As a consequence of the high amount of EGFR in BT-20, it is much more probable that EGF activation leads to EGFR homodimerization and renders the observation of MET diffusing slower due to heterodimerization or interactions with EGFR less likely than in HeLa cells where both receptors are expressed in similar proportions. This last fact also shows the complementarity of the applied single-molecule methods as receptor densities determined by DNA-PAINT, where most MET receptors on the cell membrane are visualized, revealed an influence of EGF activation on MET receptor densities. Taken together, our results suggest that the formation of heteromeric complexes of EGFR responds to ligand stimulation as well as receptor expression ratios. An abundance of EGFR compared to MET in BT-20 cells renders receptor interactions more probable, especially in the case of HGF-activated MET.

### 3.3. Single-Molecule Localization Techniques Offer a Systematic and Highly Sensitive Approach to Study the Subtle Phenomenon of Receptor Cross-Interaction

Single-molecule imaging tools are well suited to detect the subtle and infrequent interactions of membrane receptors. This enabled us to deliver direct proof of the formation of heteromeric MET:EGFR complexes and to discuss these in the context of ligand stimulation and receptor expression ratios. At the same time, we showed that upon application of moderate ligand concentrations, western blotting was not sensitive enough to detect any cross phosphorylation of receptors (Appendix A). Increased MET phosphorylation levels upon EGF stimulation have been detected previously by western blots, though only when applying distinctly higher ligand concentrations [18,19].

In summary, we have established an approach to visualize and quantitatively evaluate receptor cross-interactions in living and fixed cells in relation to receptor expression ratios and cell lines. This may become a valuable tool to predict the success of the application of single or of a combination of several receptor inhibitors in cancer therapy [15,45,46]. Our method is easily adaptable to other cell lines or further receptors which may also play a role in cross-interaction such as the ErbB receptor family [18,41].

## 4. Materials and Methods

### 4.1. Coverslip Passivation and Functionalization

25 mm glass coverslips (VWR International, Radnor, PA, USA) were coated with poly-L-lysine grafted polyethylene glycol (PLL-PEG) to reduce unspecific interactions of fluorophore-labeled ligands with the glass surface in SPT experiments. The surface coating was complemented with the peptide motif arginyl-glycyl-aspartic acid (RGD) attached to the PLL-PEG (70% molar ratio of RGD relative to PLL-PEG) to enable cell adhesion. For PLL-PEG-RGD synthesis, maleimide-PEG-*N*-hydroxysuccinimide (Rapp Polymere GmbH, Tübingen, Germany) was incubated with the custom-synthesized peptide Ac-CGRGDS-COOH which was a kind gift of the Tampé group (Frankfurt University, Germany) in 10 mM HEPES (Roth, Karlsruhe, Germany) buffer with 150 mM NaCl (Sigma-Aldrich, St. Louis, MO, USA) (pH 6.5) for 15 min at room temperature. PLL (Sigma-Aldrich) in 50 mM sodium borate (Sigma-Aldrich) buffer (pH 8.4) was added, the pH of the mixture was adjusted to 8, and incubated at room temperature overnight. The procedure of coverslip coating was described in Harwardt et al. (2017). Briefly, plasma-cleaned (Zepto B, Diener Electronic GmbH, Ebhausen, Germany) glass slides were incubated with 0.8 mg/mL PLL-PEG-RGD for 90 min, nitrogen-dried, and stored at −20 °C until further use.

DNA-PAINT experiments of fixed cells were conducted in 8-well chambered coverslip (Sarstedt, Nümbrecht, Germany) coated with fibronectin (Sigma-Aldrich) to facilitate cell growth on glass coverslips. Fifteen µg/mL fibronectin was incubated at 37 °C for one hour and air-dried for one additional hour under sterile conditions. Fibronectin coating was prepared shortly before cell seeding.

### 4.2. Cell Culture

Experiments were performed in HeLa (Institut für angewandte Zellkultur, Munich, Germany) and BT-20 cells (Cell Lines Service GmbH, Eppelheim, Germany). Cells were grown in high glucose Dulbecco’s modified Eagle medium (DMEM) in the case of HeLa cells or in DMEM-F12 in the case of BT-20 cells, each complemented with 10% fetal bovine serum (FBS), 1% GlutaMAX, and 100 U/mL penicillin and 100 μg/mL streptomycin at 37 °C and 5% CO_2_ in an automatic CO_2_ incubator (Model C 150; Binder GmbH, Tuttlingen, Germany) for 2 to 4 days. All cell culture chemicals were purchased from Gibco, Life Technologies (Thermo Fisher Scientific GmbH, Waltham, MA, USA).

For SPT experiments, cells were seeded onto PLL-PEG-RGD-coated coverslips (25 mm diameter) in six-well plates (Greiner, Bio-One International GmbH, Kremsmünster, Austria). Wells contained 3, 5, or 10 ∙ 10^4^ HeLa cells or 7.5 or 10 ∙ 10^4^ BT-20 cells and were grown in DMEM or DMEM-F12 with 10% FBS and 1% GlutaMAX at 37 °C and 5% CO_2_ for 4, 3, or 2 days, respectively. For control experiments, 15 ∙ 10^4^ or 20 ∙ 10^4^ HeLa cells were seeded and grown for 2 or 3 days prior to transfection. For studies of long-term effects of EGF ligands, the latter was added in serum-free medium for different periods.

For DNA-PAINT experiments, either 2 ∙ 10^4^ HeLa cells or 2.5 ∙ 10^4^ BT-20 cells per well were seeded in fibronectin-coated coverslips. They were incubated in DMEM or DMEM-F12 respectively with 10% FBS and 1% GlutaMAX at 37 °C and 5% CO_2_ for one day.

### 4.3. Single-Particle Tracking

#### 4.3.1. Sample Preparation

The antibody fragment 3H3-Fab was a kind gift from Hartmut H. Niemann (Bielefeld University, Germany). The labeling procedure with ATTO 647N-N-hydroxysuccinimide ester (ATTO-TEC, Siegen, Germany) is described in Harwardt et al. (2017). Purified Fab-ATTO 647N was obtained in a concentration of 5.05 µM and a degree of labeling of 168%.

EGFR-SOMAmer [33] reagent was modified with a DNA docking strand P1 (5′ TTATACATCTA 3′) for PAINT imaging [47]. Both ligands were stored at −20 °C in protein or DNA LoBind tubes (Eppendorf GmbH, Hamburg, Germany), respectively.

As a negative control, a model transmembrane protein consisting of an extracellular monomeric enhanced green fluorescent protein (mEGFP) and an artificial transmembrane domain (TMD) was used [35]. The pSems-mEGFP-TMD plasmid was a kind gift from Jacob Piehler (University of Osnabrück, Germany). Cells were transfected with 250 ng/well plasmid and 2.25 µg/well salmon DNA (Invitrogen, Thermo Fisher Scientific GmbH, Waltham, MA, USA) using Lipofectamine 3000 (Thermo Fisher Scientific GmbH, Waltham, MA, USA). Transfected cells were incubated at 37 °C and 5% CO_2_ for 16–24 h.

Shortly before SPT measurements, coverslips with adherent cells were transferred to custom-built coverslip holders and rinsed once with 600 µL of imaging buffer (IM) consisting of DMEM with 1% GlutaMAX and 50 mM HEPES (Gibco, Life Technologies). Cells were cooled to room temperature in fresh 600 µL of IM for 15 min. For tracking MET, Fab-ATTO 647N was added to the sample with a final concentration of 0.25 nM, optionally 1 nM HGF (PeproTech GmbH, Hamburg, Germany) or 100 ng/mL EGF (Sigma-Aldrich) were added. For SPT experiments on EGFR, SOMAmer reagents were heat-cooled in IM at 90 °C for 5 min and then cooled to room temperature for approximately 20 min. Cell samples were incubated with 100 nM SOMAmer reagent solution plus 10 µM Z-Block for quenching of unspecific binding events [33] for 10 min at room temperature. After rinsing two times with 600 µL IM, the labeled DNA imager strand P1-Cy3B (Eurofins Genomics, Luxembourg, Luxembourg) was added to the sample (10 nM in 600 µL IM). For tracking TMD, 1 nM FluoTag^®^-Q nanobody (Nanotag Biotechnologies GmbH, Göttingen, Germany) targeting mEGFP and labeled with Abberior Star 635P were added to the sample. If needed, HGF or EGF were added in the same concentrations as for SPT experiments on MET.

#### 4.3.2. Setup and Data Acquisition

Data were recorded with an N-STORM microscope (Nikon, Düsseldorf, Germany) applying a 647 nm laser for excitation of ATTO 647N or Abberior Star 635P or a 561 nm laser for excitation of Cy3B, both in total internal reflection fluorescence (TIRF) illumination mode. For experiments that served as negative control, transfected cells expressing mEGFP-TMD were identified by the fluorescence signal of mEGFP which was excited with a 488 nm laser in TIRF mode. The software tools Micro-Manager (version 1.4.14) [48] and NIS Elements (version 4.30.02, Nikon, Düsseldorf, Germany) were used for measurement control. The objective (100x Apo TIRF oil) featured a 100× magnification and a numerical aperture of 1.49. Laser intensities were adjusted to 0.2 kW/cm^2^ for the 647 nm laser and to 0.05 kW/cm^2^ for the 561 nm laser. The following camera (DU-897U-CS0-#BV, Andor Technology, Belfast, UK) settings were applied: a frame time of 20 ms, 1000 frames per movie, an image size of 256 × 256 pixels, an EM gain of 200 for measurements with ATTO 647N or Abberior Star 635P and of 300 for Cy3B, and an activated frame transfer. Measurements of one coverslip never exceeded a total time span of 30 min. A reference sample without HGF and EGF was imaged either before or after each sample with ligands. 50 cells per condition were measured for analysis of MET and EGFR dynamics in non-transfected cells. In cells transfected with mEGFP-TMD, 46–47 (or 38–46) cells were analyzed and the diffusion dynamics of mEGFP-TMD (or MET) was determined. All presented SPT data were collected on at least three separate measurement days using three independent cell batches.

#### 4.3.3. Data Analysis

SPT data were analyzed using PALM-Tracer (Bordeaux Imaging Center, France), a plugin for MetaMorph (version 7.7.0.0, Molecular Devices, Sunnyvale, CA, USA). Parameter settings and the analysis process were described previously [38]. Single molecules were localized by centroid fitting, and MSD values and diffusion coefficients for single trajectories were calculated.

#### 4.3.4. Normalized Diffusion Coefficients and Diffusion Type Analysis

Mean diffusion coefficients were calculated for each condition from 50 cells for MET and EGFR in wildtype cells, from 46–47 cells for mEGFP-TMD in transfected cells, and from 38–46 cells for MET in transfected cells. To alleviate the influence of small temperature fluctuations or daily variations of cellular states, these mean diffusion coefficients were normalized by calculating a relative value for ligand treated cells compared to reference measurements of ligand-free samples recorded either directly before or after the respective condition. Normalized diffusion coefficients were depicted in box plots (Figure 3b,e and Appendix A). Normalized diffusion coefficients determined after long-term incubation of EGF were depicted as dot plot against the ligand incubation time (Appendix A).

To obtain a dynamic fingerprint of MET and EGFR after exposure to HGF and EGF, we performed a diffusion type analysis and determined the proportions of immobile, confined, and freely diffusing subpopulations as well as the respective mean diffusion coefficients following a published procedure [38]. Briefly, the dynamic localization precision was determined [49] from nine tracking movies recorded on various measurement days. Based on this value, the lowest determinable diffusion coefficient was calculated amounting to 3.7 ∙ 10^−3^ µm^2^/s. This value was combined with the method published by 50) to determine diffusion types. Trajectories yielding a diffusion coefficient lower than 3.7 ∙ 10^−3^ µm^2^/s were assigned as immobile. Confined and free diffusion were discriminated according to the Rossier method [50] applying a τ value of 120 ms as threshold. The relative occurrences and mean diffusion coefficients of the three diffusive subpopulations were represented as bar plots (Appendix A).

### 4.4. Exchange-PAINT

#### 4.4.1. Sample Preparation

Secondary antibodies were labeled with DNA docking strands via maleimide-PEG_2_-succinmidyl ester (Sigma-Aldrich) as a molecular linker according to a published protocol (30). Briefly, thiolated DNA (Eurofins Genomics) was reduced by incubation with 250 mM dithiothreitol (Sigma-Aldrich) for 2 h at room temperature. Crosslinker and antibody were incubated in a 10:1 molar ratio for 90 min at 4 °C. Reduced DNA and crosslinked antibody were mixed in a 10:1 molar ratio and incubated overnight at 4 °C slightly rotating. Excess DNA was eliminated via 100 kDa Amicon spin filters (Merck, Darmstadt, Germany). Secondary antibodies were modified with DNA docking strands: donkey anti-goat IgG (Jackson ImmunoResearch, West Grove, PA, USA, # 705-005-003) with P5 (5′ TTTCAATGTAT 3′) for staining MET and rabbit anti-mouse IgG (Jackson ImmunoResearch; # 315-005-003) with the DNA docking strand P1 (5′ TTATACATCTA 3′) for staining EGFR [47]. Labeled antibodies were stored at 4 °C.

Cells seeded on 8-well plates were starved for approximately 2 h in a serum-free DMEM medium. For stimulation, either 1 nM HGF or 100 ng/mL EGF were added in serum-free medium to the samples and incubated for 10 min at 37 °C. Samples were rinsed once with 0.4 M sucrose (Sigma-Aldrich) in phosphate-buffered saline (PBS), pH 7.4 (prepared from 10× concentrate, product number D1408, Sigma-Aldrich), and were subsequently incubated with fixation solution for 15 min at room temperature consisting of 4% methanol-free formaldehyde (Sigma-Aldrich), 0.1% glutaraldehyde (Sigma-Aldrich), and 0.4 M sucrose in PBS. Cells were rinsed three times with PBS and next incubated with the blocking buffer containing 5% (*w*/*v*) bovine serum albumin (BSA) (Sigma-Aldrich) in PBS for one hour at room temperature. Samples were incubated with the primary antibodies in blocking buffer while shaking slowly at room temperature for two hours. 2 µg/mL primary goat anti-human MET IgG (R&D Systems, Minneapolis, MN USA, AF276) and 4µg/mL mouse anti-EGFR IgG (Santa Cruz Biotechnology, Dallas, TX, USA, sc-120) were used. Samples were rinsed three times with PBS and incubated with the DNA-labeled secondary antibodies, either 50 µg/mL donkey anti-goat IgG-P5 or 50 µg/mL rabbit anti-mouse IgG-P1, in blocking buffer shaking slowly at room temperature for one hour. Cells were rinsed three times with PBS and subsequently post-fixated with 4% methanol-free formaldehyde in PBS for 10 min at room temperature. After rinsing three times with PBS, samples were either processed for measurements or stored in 0.01% NaN_3_ (Roth) in PBS at 4 °C for a maximum of two days.

For alignment of the MET and EGFR channel in image post-processing, fiducial markers were used. A solution of 90 nm gold beads (Nanopartz, Loveland, CO, USA) was sonicated for 10 min, diluted 1:5 in PBS and sonicated once more for 10 min. 300 µL of the gold bead dilution was added to each well. After 10 min, the samples were rinsed three times with PBS. P1 (5′ CTAGATGTAT 3′) and P5 (5′ CATACATTGA 3′) DNA imager strands [47] labeled with ATTO 655 (bought as labeled constructs from Eurofins Genomics) were diluted in imaging buffer containing 0.5 M NaCl (Sigma-Aldrich) in PBS to a final concentration of 2 nM shortly before measurements.

#### 4.4.2. Setup and Data Acquisition

DNA-PAINT experiments were conducted on the N-STORM microscope. The 647 nm laser was applied in TIRF mode for excitation of ATTO 655 with a laser power of 0.6 kW/cm^2^. The following parameters for data acquisition were used: a frame time of 150 ms, 20,000 frames per movie, an image size of 256 × 256 pixels, an EM gain of 150, and an activated frame transfer. After the addition of either 300 µL P1-ATTO 655 or P5-ATTO 655 in imaging buffer to the sample. After finishing measurements with one imager strand, the sample was rinsed five times with 300 µL PBS and the second imager strand was added. The same cells were then imaged with the second imager strand. The sequence of measurements for P1 and P5 was varied. Data for each condition were acquired on at least three measurement days and three different cell batches. Six to seven cells per condition were imaged and analyzed.

#### 4.4.3. DNA-PAINT Image Processing and Receptor Density Analysis

DNA-PAINT images were processed with the software Picasso (version 2.8) [29]. First, single-molecule localizations were detected and fitted with Picasso Localize applying the following parameters: a baseline of 175 photons, a sensitivity of 4.78, a quantum efficiency of 0.9, and a min net gradient of 47,000 for MET and of 50,000 for EGFR. Super-resolved images were reconstructed in Picasso Render. A drift correction was conducted and MET and EGFR channels were aligned using the positions of fiducial markers. Localizations were linked within a radius corresponding to the nearest neighbor-based analysis (NeNA) localization precision value (approximately 10–15 nm) [51] and when exhibiting a maximum of 10 transient dark frames. Subsequently, a DBSCAN analysis [30] was performed to identify receptor clusters. The given radius corresponded to the NeNA value and the minimum localizations per cluster were set to 10 for MET and 15 for EGFR. As a final step, localizations due to unspecific binding or originating from fiducial markers were filtered out by determining an interval within the borders of the mean frame time plus or minus two times the standard deviation and only accepting localizations with a frame time within this interval. Images of receptors were generated from these processed localization lists.

Filtered images were loaded into Fiji (NIH, USA) [52]. The area of each cell was marked and measured. The number of objects representing the amount of receptor clusters on the membrane within the set cell area were counted with the plugin “3D object counter” [53]. The number of receptors per µm^2^ of cell membrane was calculated for 5 cells per condition and averaged. Receptor densities of MET and EGFR in HeLa and BT-20 cells in unstimulated and ligand-activated states are represented as bar plots with corresponding standard deviations (Figure 1c,d). Additionally, a schematic illustration of relative receptor numbers was designed.

#### 4.4.4. Coordinate-Based Colocalization Analysis of MET and EGFR

Coordinate-based colocalization (CBC) analysis [31] on MET and EGFR was performed in the LocAlization Microscopy Analyzer (LAMA, version 16.10) software [32] using camera parameters as described above. Processed and filtered Picasso localization lists were converted into the LAMA text file format. The CBC analysis was run on batch files of EGFR and MET filtered DBSCAN data of the same cells with a set radius of 30 nm and three increments. CBC values ranging between −1 and 1 were determined for each cluster either using MET or EGFR as a point of origin. Positive CBC values indicate colocalization, negative values represent segregated but close localizations, a CBC value of 0 represents no colocalization. CBC results were filtered for values of 0.15–1 representing colocalized receptor clusters. CBC-based images were generated from these filtered localization lists (Figure 2b). The number of colocalizing clusters were determined from these images within the same ROIs as applied for the calculation of total receptor numbers using the 3D object counter tool in Fiji. This also ensured that remaining unspecific signals on the glass surface or dense regions at cell margins were excluded from the colocalization analysis. The relative amount of colocalizing clusters in comparison to the total amount of either MET or EGFR in HeLa and BT-20 cells under all stimulation conditions was calculated. Results were averaged over 5 to 7 cells per condition and represented in bar plots with respective standard deviations (Figure 2c). Additionally, densities of co-localizing MET:EGFR clusters were depicted relative to MET and EGFR densities in Venn diagrams (Figure 2d). For this, the numbers of co-localizing receptor clusters determined in the MET and EGFR channel were averaged. In general, numbers of co-localizing clusters were very similar in both channels. These averaged values were divided by the respective cell area to obtain cluster densities. Venn diagrams were generated using the online tool meta-chart.

### 4.5. Statistical Analysis

Two-sample t-tests were performed in OriginPro (version 2016G, Origin Lab Corporation, Northampton, MA, USA) to test the significance of differences between the chosen conditions for receptor densities determined from DNA-PAINT data (Figure 1c,d) and for normalized diffusion coefficients (Figure 3b,e). Resulting *p*-values were categorized according to level of significance as follows: *p* > 0.05 no significant difference between populations (n.s.), *p* < 0.05 significant difference (*), *p* < 0.01 very significant difference (**), *p* < 0.001 highly significant difference (***). In addition, for normalized diffusion coefficients, the distribution of mean differences was calculated according to Ho et al. (2019)) for each condition in comparison to the unstimulated sample. The 95% confidence interval of the mean difference distribution was indicated to illustrate the significance.

### 4.6. Western Blot Analysis

#### 4.6.1. Ligand Stimulation and Cell Lysis

Cells were seeded on 10 cm cell culture dishes (Greiner, Bio-One International GmbH) with densities of 100 ∙ 10^4^ cells/dish for HeLa cells and 200 ∙ 10^4^ BT-20 cells/dish. They were incubated in DMEM or DMEM-F12 with 10% FBS and 1% GlutaMAX for three days at 37 °C with 5% CO_2_. After incubating cells in serum-free media overnight, cells were stimulated with one ligand or a combination of ligands in the same concentrations as applied in SPT experiments in serum-free DMEM with 1% GlutaMAX at 37 °C and 5% CO_2_ for 5 to 10 min. Subsequently, cells were rinsed with 10 mL of ice-cold PBS and dishes were kept on ice for at least two minutes. Ice-cold lysis buffer, consisting of 50 mM Tris (Sigma-Aldrich), 150 mM NaCl, 1% Triton X-100 (Sigma-Aldrich), 1mM Na_3_VO_4_ (Sigma-Aldrich), 1 mM EDTA (Sigma-Aldrich), 1mM NaF (Sigma-Aldrich), and one-fourth of a cOmplete Mini EDTA-free protease inhibitor tablet (Roche, Basel, Switzerland), was added drop-wise to the sample, and cells were scraped from the dish, collected in cooled Eppendorf tubes, and shaken at 750 rpm at 4 °C for 5 min (Thermo-Shaker, Universal Labortechnik GmbH & Co. KG, Leipzig, Germany). Samples were centrifuged at 12,000 rpm at 4 °C for 20 min (Centrifuge 5418 R, Eppendorf), pellets were discarded while the supernatant was collected for further analysis.

#### 4.6.2. SDS-PAGE Gel Electrophoresis

The protein concentration of each sample was determined by using a BCA Protein Assay Kit (VWR International GmbH) and performing concentration measurements at a nanophotometer (Implen GmbH, München, Germany). For analysis of the influence of labeled ligands on MET and EGFR stimulation, a sodium dodecyl sulfate-polyacrylamide gel (SDS-PAGE) was prepared by mixing 1.875 mL of 1.5 M Tris-HCl (pH 8.8), 37.5 µL of 20% (*w*/*v*) SDS, 1.5 mL of 40% (*w*/*v*) acrylamide/ bisacrylamide, 37.5 µL of 10% (*w*/*v*) ammonium persulfate (APS), and 5 µL of tetramethylethylenediamine (TEMED) in water (all purchased from Sigma-Aldrich). After 30 min, this separation gel was solidified and the collection gel was added by mixing 371 µL of 1.5 M Tris-HCl (pH 6.8), 7.5 µL of 20% (*w*/*v*) SDS, 161 µL of 40% (*w*/*v*) acrylamide/ bisacrylamide, 7.5 µL of 10% (*w*/*v*) APS, and 1.5 µL of TEMED in water. After further two hours of incubation, the gel was ready for electrophoresis. For possible detection of cross-phosphorylation of receptors, 4–20% gradient SDS-PAGE gels (BioRad Laboratories, Hercules, CA, USA) were used. Gels were mounted in a cask filled with running buffer containing 3 g Tris-HCl, 14.4 g glycine (Sigma-Aldrich), and 1 g SDS per 1 L of water. Each sample containing 50 µg of protein were mixed with 20% (*v*/*v*) SDS loading dye (250 mM Tris-HCl (pH 6.8), 8% (w/v) SDS, 0.1% (*w*/*v*) bromophenol blue (Sigma-Aldrich), 40% (*v*/*v*) glycerol (Roth) in water), and 10% (*v*/*v*) 1M dithiothreitol. Prepared samples were heated to 95 °C for 5 min and loaded onto the gel. A page ruler (Thermo Fisher Scientific) served as a reference marker. Each gel was run at 100 V for 90 min.

#### 4.6.3. Western Blots

Western blots were obtained from gels within 7 min by using an iBlot Gel Transfer System (Invitrogen). Blots were incubated with blocking buffer containing 5% (*w*/*v*) milk powder (nonfat dry milk, Cell Signaling Technology, Danvers, MA, USA) in TBST (1.2 g Tris(hydroxymethyl)aminomethane (Sigma-Aldrich), 4.4 g NaCl, 0.05% (*v*/*v*) Tween-20 (Sigma-Aldrich) in 400 mL water) shaking at room temperature for one hour. After rinsing three times with TBST for 5 min each, primary antibodies (EGF Receptor antibody (Cell Signaling Technology, #2232), Phospho-EGF Receptor (Tyr1068) antibody (Cell Signaling Technology, #2234), Met antibody (Cell Signaling Technology, #4560), Phospho-Met (tyr1234/1235) (D26) XP rabbit antibody (Cell Signaling Technology, #3077)) were incubated on the blots in TBST supplemented with 5% (*w*/*v*) BSA shaking at 4 °C overnight. At the same time, actin was labeled as a control with a primary rabbit anti-actin antibody (abcam, Cambridge, UK, ab14130). Blots were rinsed three times with TBST at room temperature for 5 min each, before incubation with the secondary HRP-tagged antibody (goat anti-rabbit IgG, Jackson ImmunoResearch) diluted 1:20,000 in TBST with 5% (*w*/*v*) BSA shaking at room temperature for three hours. Blots were rinsed four times with TBST for 15 min each and once with TBS for 5 min. For band detection, blots for the analysis of the influence of labeled ligands were incubated with SIGMAFAST diaminobenzidine tablets with a metal enhancer (Sigma-Aldrich) dissolved in 5 mL of water at room temperature shaking gently for 5 min. Western blots were rinsed with water and air-dried. For a more sensitive detection in the context of possible cross-phosphorylation of receptors, the respective blots were incubated with 1 mL of SuperSignal West Femto Maximum Sensitivity Substrate (Thermo Fisher Scientific) solution for approximately 1 min. Bands were visualized via chemiluminescence detection at a Fusion FX Edge imager (Vilber Lourmat, Collégien, France) with a set illumination time of 2 s.

## Figures and Tables

**Figure 1 ijms-21-02803-f001:**
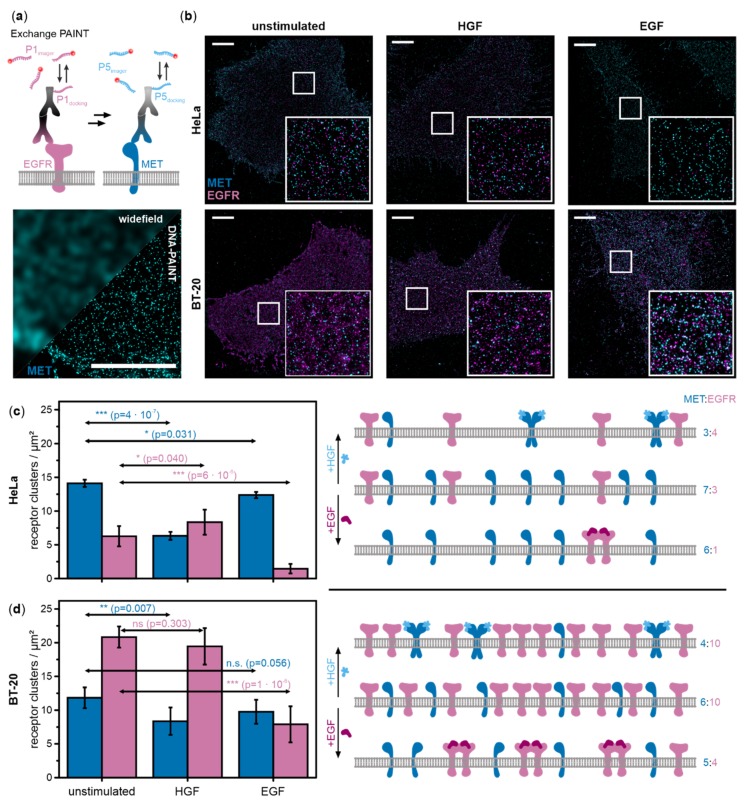
MET and epidermal growth factor receptor (EGFR) densities in the cell membrane of HeLa and BT-20 cells. (**a**) Concept of super-resolution microscopy by Exchange-PAINT of MET and EGFR (top). A super-resolved image is obtained from transient binding events of short, fluorescently labeled DNA oligonucleotides to DNA-labeled antibodies. The image (bottom) shows MET visualized by widefield microscopy versus DNA-PAINT (scale bar 5 µm). (**b**) Exchange-PAINT images of MET (cyan) and EGFR (magenta) immunostained with secondary antibodies carrying P5 and P1 DNA docking strands and labeled with complementary DNA imager strands labeled with ATTO 655. Total internal reflection fluorescence (TIRF) images of the plasma membrane of HeLa and BT-20 cells were recorded either in the unstimulated state, after hepatocycte growth factor (HGF) stimulation, or after activation with epidermal growth factor (EGF) (scale bar 5 µm, insets 5 µm × 5 µm). Receptor cluster densities of MET (cyan) and EGFR (magenta) in (**c**) HeLa and (**d**) BT-20 cells were determined from DNA-PAINT images (*n* = 6–7 cells/condition from at least three independent experiments) and plotted in the histogram (left). (Note that receptor clusters refer to both monomers and dimers.) Error bars represent standard deviations. Results of two-sample t-tests for comparison of activated samples with the respective unstimulated sample are depicted as arrows (*p* > 0.05 no significant difference between populations (n.s.), *p* < 0.05 significant difference (*), *p* < 0.01 very significant difference (**), *p* < 0.001 highly significant difference (***)). The quantitative data was used to generate density and activation schemes of MET and EGFR in HeLa and BT-20 (numbers at the right indicate relative receptor ratios at the cell membrane determined from DNA-PAINT images).

**Figure 2 ijms-21-02803-f002:**
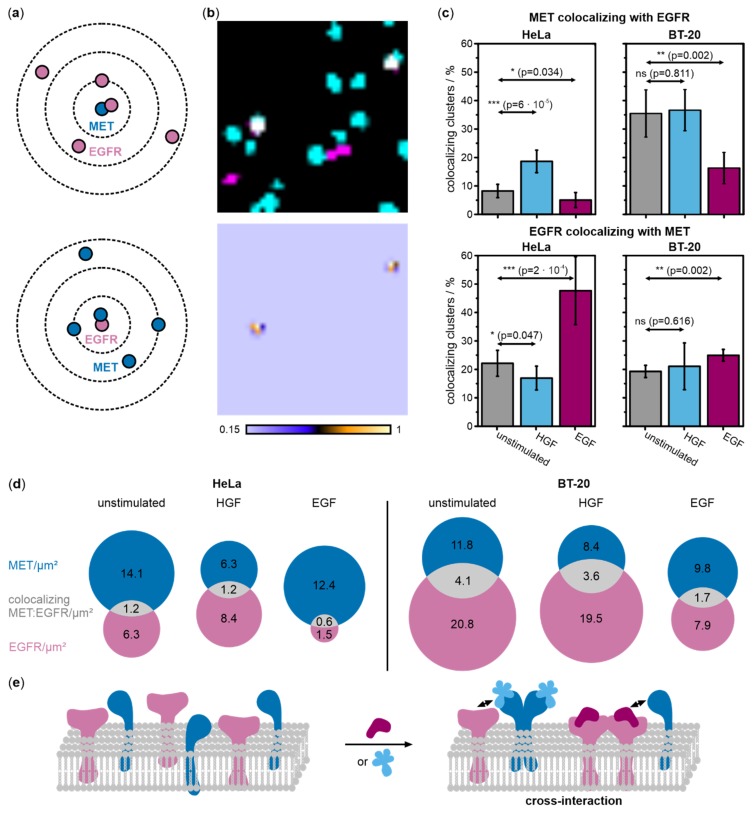
Coordinate-based colocalization analysis of DNA-PAINT data of MET and EGFR reveals increased receptor colocalization upon ligand stimulation in HeLa and BT-20 cells. (**a**) Representation of the principle of coordinate-based colocalization (CBC) of MET (cyan) and EGFR (magenta). For each receptor localization, the numbers and distances of surrounding receptors of the other species were determined (search radius *r* = 30 nm) and a distribution function calculated that reports on colocalization (−1 < CBC > 1). (**b**) Dual-color super-resolution images of MET and EGFR (top) were transformed into colocalization images (bottom) (0.15 < CBC < 1) (image sizes are 1 µm × 1 µm). (**c**) The relative amount of MET and EGFR colocalizing in single clusters in HeLa and BT-20 cells with respect to the total amount of the respective receptor in unstimulated (grey), HGF-activated (light blue), and EGF-stimulated (purple) cells. Values were averaged over 5 to 7 cells from at least three independent experiments. Error bars represent standard deviations. Results of two-sample *t*-tests for comparison of activated samples with the respective unstimulated sample are depicted as arrows (*p* > 0.05 no significant difference between populations (n.s.), *p* < 0.05 significant difference (*), *p* < 0.01 very significant difference (**), *p* < 0.001 highly significant difference (***)). (**d**) Receptor cluster densities (per µm^2^) on the cell membrane of MET (cyan) and EGFR (magenta) together with colocalizing MET:EGFR clusters (gray) shown as Venn diagrams for HeLa and BT20 cells. Densities of co-localizing receptor clusters were calculated from an average of the number of co-localizing clusters in the MET and EGFR channel (see Materials and Methods). (**e**) A model of MET and EGFR cross-interaction upon stimulation with either EGF or HGF.

**Figure 3 ijms-21-02803-f003:**
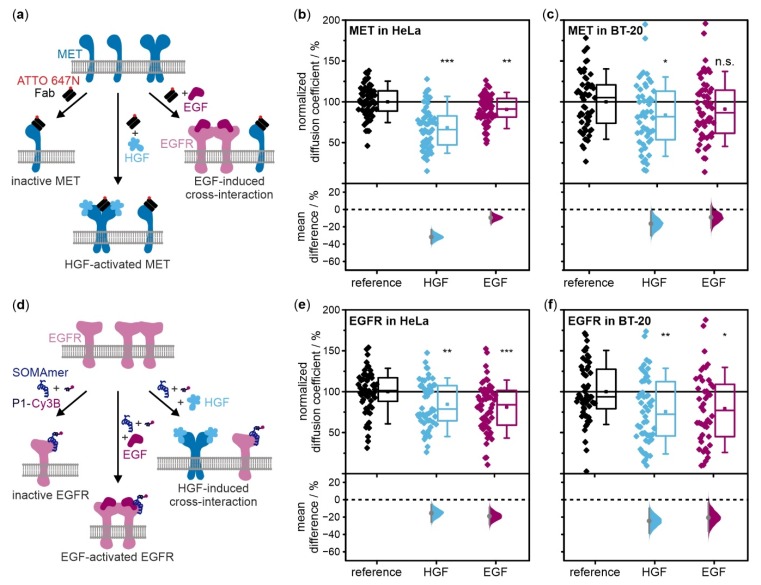
Diffusion dynamics of MET and EGFR in HeLa and BT-20 cells after stimulation with HGF and EGF studied by SPT indicate receptor cross-interactions. (**a**,**d**) MET was tracked by adding Fab-ATTO 647N. EGFR was tracked by adding EGFR-SOMAmer reagent modified with a P1-docking strand and P1-Cy3B for PAINT imaging. The Fab fragment and the SOMAmer reagent bind to the respective receptor but do not activate them. For observations of direct activation or possible cross-interaction, unlabeled HGF (light blue) or EGF (purple) were added to the samples. Diffusion coefficients of MET and EGFR in resting and activated cells (each *n* = 50 from at least three independent experiments) were determined in (**b**,**e**) HeLa and (**c**,**f**) BT-20 cells. All diffusion coefficients were normalized against reference measurements of ligand-untreated cells for all types of treatment. The box plots of diffusion coefficients display the 5th percentile, 25th percentile, median (line), mean (square), 75th percentile, and 95th percentile. Results of two-sample t-tests for comparison of ligand-treated cells with the reference are depicted above the box plots (*p* > 0.05 no significant difference between populations (n.s.), *p* < 0.05 significant difference (*), *p* < 0.01 very significant difference (**), *p* < 0.001 highly significant difference (***)). The lower y-axes of the graphs depict the distribution of mean differences [34] of each condition in comparison to the unstimulated sample. The mean difference is represented as a grey dot; each 95% confidence interval of the mean difference distribution is indicated by vertical grey error bars.

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
