# Peer review of "Single-Molecule Super-Resolution Microscopy Reveals Heteromeric Complexes of MET and EGFR upon Ligand Activation"

_ijms, 2020, doi:10.3390/ijms21082803_

Round 1

Reviewer 1 Report

In this manuscript entitled "Single-molecule super-resolution microscopy reveals heteromeric complexes of MET and EGFR upon ligand activation", authors presented static and dynamic evidence of the presence of heteromeric clusters of MET and EGFR on the cell membrane of two cancer cell lines. This work is interesting as it will open the door for further functional studies to be done on other receptors to understand the impact of alternative mechanisms of different signaling pathways.

There are few technical comments:

Western blot is not the best way to study protein-protein interaction. Phosphorylation of the receptors could transient. I recommends the author to perform co-immunoprecipitation experiment using the same experimental conditions. The other recommendation is look into downstream target genes of these two receptors and whether it will be initiated under the stimulation by HGF and/or EGF. I suggests the the authors to look into another receptor that could serve as a negative control (to exclude the possibility of nonspecific effect). Further manipulation of the cellular conditions will add significant value to solidify the main results. For instance, knockdown or overexpression of either or both receptors will substantiate the main results.  

I have the following minor language/style comments:

Line 20, BT20 should be replaced with BT-20. Line 67, the reference number 24 should be replaced with "Park and his colleagues" or " Park et al". Point 2.1 should be italic. Line 224, h should be replaced with hours, Line 353, see point 2 in these comments.

Reviewer 2 Report

This study presents single molecule super-resolution analysis of complexes between EGF receptor (EGFR) and Met receptor (MET) in two different cell types with different ratios of MET:EGFR expression. It has been known for some time that trans-activation of these two receptors (e.g. leading to MET phosphorylation when cells are incubated with EGFR ligand and vice versa) can occur under certain conditions but the underlying molecular mechanisms have remained unclear. The study is thus timely and adds to our understanding of receptor cross-talk.

The presented data show stimulation with their own ligands leads to reductions in MET and EGFR cell membrane densities in both HeLa and BT-20 cells. Coordinate-based colocalization analysis showed an increase in co-localization of both receptors when they were stimulated with their own ligands. Using single-particle tracking, the authors also show ligand-induced reduction of diffusion.

The manuscript is well written with an extensive Method section that gives a lot of experimental detail, and the data are generally well presented and support the conclusions. However, I am not clear on how many independent experiments the data are based on. Below are my suggestions for amendments.

  1. Image selection. The image data are generally based on the analysis of 5 cells. It is not clear to me what was done to minimize experimental bias. Was the study carried out in a blinded fashion? How were the 5 cells chosen on which the data analysis was carried out?
  2. I cannot find any information on how many independent experiments were carried out for the various experiments. If the data are from a single experimental repeat, they are not valid. Please repeat the experiments in such a case and state the number of independent experiments for each type of experiment.
  3. The graphical depictions are helpful for the reader but could be improved to better represent the data. Figs 1c and 1d: Please adjust the number of receptor molecules a bit better to the actual receptor density (e.g. MET:EGFR 14:7 ratio is shown with 8:5 molecules; 12:2 ratio with 7:2 etc).
  4. It is not clear how the co-localized fractions were calculated in the Venn diagram in Fig 2d; please add this information to the legend (whether based on MET co-localizing with EGFR or EGFR co-localizing with MET).
  5. The diagram in Fig 2e is somewhat misleading, as the manuscript does not include any data that show actual formation of heterodimers between EGFR and MET. Please remove the cross-activation diagram, as it is not clear what this diagram is supposed to show (the term ‘cross-activation’ is normally linked to kinase activation).
  6. Figure 3. Please add the distribution of the data for the unstimulated cells to the lower plots in Figures 3b and 3c, so the reader can see the distribution of the data for unstimulated cells (may have to change the y axis for this).
  7. The schematics in Figures 3a and 3d are somewhat misleading, as they seem to suggest that fluorescent labels bind to the same sites as the natural ligands. The data show no influence of the fluorescent ligands on receptor activation by the fluorescent ligands, hence suggesting that they bind to different sites than he natural ligands. Please reflect different binding sites for physiological and fluorescent ligands in the graphics.
  8. The signal detection for the Western blots was carried out with a method that is very crude and does not allow sensitive detection of phospho-tyrosine signals. Please repeat the experiments with much more sensitive detection (under conditions where the signals are not saturated), using either ECL or fluorescent secondary antibodies and a state-of-the art digital scanner. Performing the detection in this way, may well show trans-phosphorylation signals.
  9. Figure S4. The EGFR blot from HeLa cells has been inappropriately manipulated with the marker lane spliced in from a different blot/different part of the blot. Please indicate clearly where the data are not from the same blot.
  10. The statement “Western blots are a well-established method to inform on protein interactions.” is not correct as such. Western blots can detect individual proteins (or phospho-forms) on a membrane but not ‘protein interactions’. Please re-write the statement. Similar statements that are made at the beginning of the Discussion should be amended as well.

Round 2

Reviewer 2 Report

The authors have done a careful revision and addressed all my points to my satisfaction.